

# Confirmatory digital subtraction angiography after clinical brain death/ death by neurological criteria: impact on number of donors and organ transplants

Karen Irgens Tanderup Hansen[1,2,*], Jesper Kelsen[3,*], Marwan H. Othman[2], Trine Stavngaard[3] and Daniel Kondziella[2,4]

[1] University of Southern Denmark, Faculty of Health Science, Odense, Denmark
[2] Department of Neurology, Rigshospitalet, Copenhagen University Hospital, Copenhagen, Denmark
[3] Department of Radiology, Rigshospitalet, Copenhagen University Hospital, Copenhagen, Denmark
[4] Department of Clinical Medicine, University of Copenhagen, Copenhagen, Denmark
* These authors contributed equally to this work.

## ABSTRACT

**Background:** Demand for organs exceeds the number of transplants available, underscoring the need to optimize organ donation procedures. However, protocols for determining brain death (BD)/death by neurological criteria (DNC) vary considerably worldwide. In Denmark, digital subtraction angiography (DSA) is the only legally approved confirmatory test for diagnosing BD/DNC. We investigated the effect of the time delay caused by (repeat) confirmatory DSA on the number of organs donated by patients meeting clinical criteria for BD/DNC. We hypothesized that, first, patients investigated with ≥2 DSAs donate fewer organs than those investigated with a single DSA; second, radiological interpretation of DSA is subject to interrater variability; and third, residual intracranial circulation is inversely correlated with inotropic blood pressure support.

**Methods:** All DSAs performed over a 7-year period as part of BD/DNC protocols at Rigshospitalet, Copenhagen University Hospital, Denmark, were included. Clinical data were extracted from electronic health records. DSAs were reinterpreted by an independent neurinterventionist blinded to the original radiological reports.

**Results:** We identified 130 DSAs in 100 eligible patients. Patients with ≥2 DSAs ($n = 20$) donated fewer organs (1.7 +/− 1.6 SD) than patients undergoing a single DSA ($n = 80$, 2.6 +/− 1.7 organs, $p = 0.03$), and they became less often donors ($n = 12$, 60%) than patients with just 1 DSA ($n = 65$, 81.3%; $p = 0.04$). Interrater agreement of radiological DSA interpretation was 88.5% (Cohen's kappa = 0.76). Patients with self-maintained blood pressure had more often residual intracranial circulation ($n = 13/26$, 50%) than patients requiring inotropic support ($n = 14/74$, 18.9%; OR = 0.23, 95% CI [0.09–0.61]; $p = 0.002$).

**Discussion:** In potential donors who fulfill clinical BD/DNC criteria, delays caused by repetition of confirmatory DSA result in lost donors and organ transplants. Self-maintained blood pressure at the time of clinical BD/DNC increases the odds for residual intracranial circulation, creating diagnostic uncertainty because radiological

Corresponding author
Daniel Kondziella,
daniel_kondziella@yahoo.com

DSA interpretation is not uniform. We suggest that avoiding unnecessary repetition of confirmatory investigations like DSA may result in more organs donated.

## INTRODUCTION

Brain death (BD), or death by neurological criteria (DNC), is a diagnosis characterized by complete and irreversible loss of brain function (*Wijdicks, 2015*). Two distinct interpretations of BD/DNC, *i.e.*, whole brain death and brainstem death, have contributed to different practices and protocols in BD/DNC determination around the world (*Lewis et al., 2020*; *Smith, 2015*).

Ancillary testing is useful (and often required by law and/or local protocols) for declaring BD/DNC when there are confounding factors that could interfere with the clinical neurological evaluation (*Wijdicks, 2015*). These include the inability to perform an apnea test due to poor oxygenation, hemodynamic instability, or evidence of chronic carbon dioxide retention.

Ancillary testing is based on the documentation of absence of electrical brain activity or cerebral blood flow. The World Brain Death Project recommends that confirmatory testing be based on cerebral blood flow measures. The hypothesis is that an increase in the intracranial pressure over the mean arterial pressure causes cessation of cerebral perfusion leading to irreversible brain damage (*Haussmann & Yilmaz, 2020*). Although non-invasive investigations are increasingly used (*MacDonald, Stewart-Perrin & Shankar, 2018*), conventional four-cerebral vessel digital subtraction angiography (DSA) is still considered the gold standard to confirm BD/DNC (*Haussmann & Yilmaz, 2020*).

Ancillary testing in BD/DNC determination is required in 28% of protocols worldwide (*Greer et al., 2020*). However, the diagnosis of BD/DNC is internationally inconsistent. Cultural, educational, and socioeconomic differences influence perceptions and practices of BD/DNC diagnostics, leading to highly variable legislations and protocols across and within countries (*Othman, Dutta & Kondziella, 2020*). Restated, confirmation of BD/DNC with ancillary testing and the tests used depend on national laws, regional practice, and local protocols reflecting individual institutions' capabilities (*Heran, Heran & Shemie, 2008*; *Citerio et al., 2014*). Also, when ancillary testing shows residual intracranial circulation, the testing is often repeated to eventually confirm absence of circulation but there is no consensus on exactly when ancillary testing should be repeated.

In 2020, 21.3% of organ donors underwent confirmatory testing in Denmark, which is in line with previous studies from other countries (24–26%) (*Sayan, 2020*; *Grzonka et al., 2023*). These studies also reveal an increased use of ancillary tests, further emphasizing the scope and importance of appropriate protocols for ancillary tests in the determination of BD/DNC.

According to Danish legislation, clinical BD/DNC protocols in adults and children over 1 year must be confirmed by DSA when the primary cause of brain injury is ischemic stroke or anoxic-ischemic encephalopathy; in case of a primary infratentorial brain lesion; or when a complete clinical BD/DNC examination is not possible (*Danish Ministry of Interior and Health, 2006*). Filling of the supraclinoid internal carotid artery above the origin of the ophthalmic artery is angiographically regarded as persistent intracranial circulation. When residual intracranial circulation is present, organ procurement is postponed until a repeated DSA shows absence of intracranial circulation (*Danish Ministry of Interior and Health, 2006*; *Kondziella, 2020*).

To identify potentials for protocol optimization, we investigated the effect of the time delay caused by confirmatory DSA on the number and quality of organ transplants donated by patients meeting clinical BD/DNC criteria. We hypothesized that patients investigated with repeated DSA donate fewer organs than those investigated with just 1 DSA. Further, we hypothesized that the radiological interpretation of DSA is subject to high interrater variability and that residual intracranial circulation is inversely correlated with the need for inotropic drugs.

## METHODS AND METHODS

We conducted a retrospective cohort study of all patients undergoing confirmatory DSA during BD/DNC determination at Rigshospitalet, Copenhagen University Hospital, a tertiary referral center, over a 7-year period from January 2016 to December 2022. Patients were identified from the institutional BD/DNC database. Clinical data were extracted from electronic health records. DSA images were anonymized and reevaluated by an independent neurointerventionalist blinded to the original reports. Inclusion criteria were patients with at least 1 DSA as part of a BD/DNC protocol. We excluded patients below the age of 1 year.

### Objectives and outcomes

The primary objective was to evaluate the effects of the time delay owing to repeated confirmatory DSAs on the frequency of organ donation, including possible negative effects on the quality and number of organ transplants. The secondary objectives were (a) to assess the interrater variability of radiological DSA interpretation; and (b) to identify clinical predictors for the presence or absence of intracranial circulation.

The primary endpoint was the number of organs donated per patient after 1 DSA compared to organs donated per patient with two or more DSAs. Secondary endpoints included (a) predictors for the absence or presence of intracranial circulation during DSA; (b) interrater variability of DSA interpretation as assessed by reevaluation by an independent blinded neurointerventional radiologist compared to the original DSA report, and (c) the quality of donated organs reflected by laboratory markers from patients with 1 DSA *vs.* patients with two or more DSAs.

## Clinical data

We collected the following clinical data: (a) baseline data including patient social security ID, sex, age, date and cause of ICU admission, as well as previous and current medical history, including treatment; (b) blood pressure at the time of clinical BD/DNC assessment, including the need for inotropic drugs, (c) radiological data including number and time of available neuroimaging (MRI, CT, DSA), DSA indication, DSA image studies, and radiological DSA reports, (d) biochemical data including specific organ markers in laboratory testing according to Scandinavian BD/DNC guidelines (*Jørgensen & Weinreich, 2023*), and (e) specific donor data including consent of organ donation and number and type of organs donated. According to Scandia Transplant, which coordinates distribution of organ transplants across the Nordic countries, certain blood biomarkers should be measured in a patient who is considered for organ donation, but many of these specific organ markers are not measured regularly (*Weinreich, 2023*). Biomarkers were available for renal (creatinine) and liver function (alanine transaminase). All study data were collected into a REDCap database complying with Danish data safety legislation.

## Statistical analysis

Data of the quality and quantity of donated organs by patients undergoing one confirmatory DSA compared to patients undergoing several DSAs were assessed by histograms and quantile-quantile-plot to check for normal distribution. As these data were not normally distributed, we used non-parametric tests for further analysis. For the primary outcome of number of organs donated, we used Wilcoxon Rank Sum test. Since patients had blood tests done at first and last DSA, a Wilcoxon Signed-Rank test was used to compare organ blood markers. We analyzed whether inotropic drugs at the time of clinical BD/DNC correlated with presence of intracranial circulation at first DSA, using a Chi-square-test and odds ratios (OR). The median contrast filling (if present) of intracranial arteries was compared between the DSA groups who received inotropic support and those who did not, using Wilcoxon Rank Sum test. For the secondary outcome of interrater variability of DSA interpretations, we used Cohen's kappa statistics. Significance was set at $p < 0.05$. All analysis were performed using R (version 2022-10-31, *R Core Team, 2022*; R-packages 'tidyverse', 'dplyr', 'ggplot2', and 'haven').

## Data availability statement

Anonymized raw data are available from the Supplemental Files. Radiological scans, including DSA, will not be shared to protect patient privacy.

## Ethics statement

The study was approved by the National Committees of Scientific Ethics in Denmark (journal number: 2213160), and next-of-kin consent was waived due to the retrospective nature of the study. Patient data were anonymized and treated according to the General Data Protection Regulation and the EU Data Protection Act.

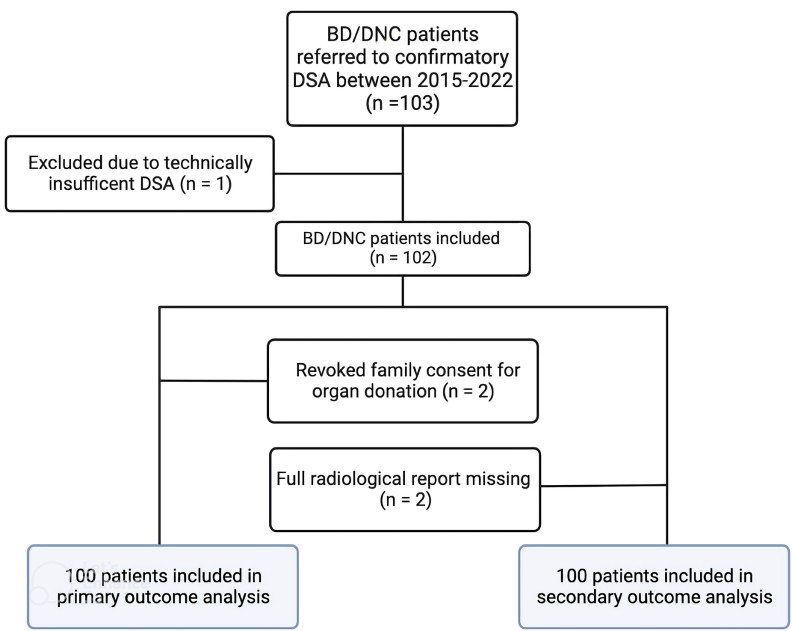

**Figure 1 Study flowchart.** This figure shows the distribution and inclusion of patients according to primary outcome (number of organs donated relative to the number of DSAs) and secondary outcomes (inotropic blood pressure support as a predictor of residual intracranial circulation, and the interrater variability of DSA interpretation).           

## RESULTS

Between January 2017 and December 2022, one hundred and three patients were referred for DSA to confirm BD/DNC. DSA was not possible due to technical difficulties with artery access in one patient, and in two patients consent to organ donation was eventually revoked by the relatives. Hence, data from 100 patients were available for primary outcome analysis. Figure 1 shows a study flow chart. Raw data are available from Table S1.

### Confirmatory DSA and number of organs donated

Eighty patients underwent 1 DSA (mean (SD) age 44.0 years (18.7), 37 women), and 20 patients underwent ≥2 DSA (42.6 years (20.1), nine women) to confirm BD/DNC diagnosis (Table 1). Ninety-one patients met clinical BD/DNC criteria according to Danish legislation (complete clinical exam was not possible in nine patients because of hemodynamic instability prevented completion of the apnea test. Reasons for referral to confirmatory DSA were as follows: ischemic stroke/post-cardiac arrest (1 DSA 76%, ≥2 DSA 75%); incomplete BD/DNC examination (1 DSA 10%, ≥2 DSA 5%); primary infratentorial lesion (1 DSA 9%, ≥2 DSA 15%); and other (1 DSA 1%, ≥2 DSA 5%). The mean number of DSA performed in patients with ≥2 DSA was 2.5. Mean time from clinical brain BD/DNC evaluation until last DSA varied between the groups (1 DSA = 211 min, ≥2 DSA = 2,422 min, $p < 0.0001$). The mean interval between repeated DSA was 945.7 min, or 15.8 h (Table 1).

   More patients with a single DSA became donors ($n = 65$ patients, 81.3%) than patients with ≥2 DSA ($n = 12$, 60%; $p = 0.044$). Similarly, patients undergoing just 1 DSA donated

Table 1 Clinical characteristics of patients, according to number of DSAs performed.

| | 1 DSA (*n* = 80) | ≥2 DSA (*n* = 20) | Significant *p*-value |
|---|---|---|---|
| **Age, mean (SD)** | 44.0 (18.7) | 42.6 (20.1) | – |
| **Sex female, *n* (%)** | 38 (47.5) | 9 (45.0) | – |
| **Previous history, number (%)** | | | |
| Hypertension | 12 (13.0) | 1 (5.0) | – |
| Hyperlipidemia | 6 (7.5) | 1 (5.0) | – |
| Atrial flutter | 2 (2.5) | 1 (5.0) | – |
| Chronic heart disease, other | 8 (10.0) | – | – |
| Chronic obstructive lung disease | 2 (2.5) | 3 (15.0) | 0.014 |
| Asthma | 8 (10.0) | 2 (10.0) | – |
| Psychiatric history | 17 (21.5) | 2 (10.0) | – |
| History of drug abuse | 6 (7.5) | – | – |
| Previous stroke | 5 (6.3) | – | – |
| Neurological diseases, other* | 7 (8.8) | 2 (10.0) | – |
| Diabetes Mellitus | 3 (3.8) | – | – |
| Cancer unspecified | 1 (1.3) | – | – |
| Other significant** | 7 (8.8) | 1 (5.0) | – |
| **Cause of admission ICU, *n* (%)** | | | |
| Cardiac arrest | 33 (41.3) | 8 (40.0) | – |
| Cardiac arrest after to suicide | 13 (16.3) | 2 (10.0) | – |
| Aorta dissection | 1 (1.3) | – | – |
| Intoxication | 4 (5.0) | 1 (5.0) | |
| Arterial ischemic stroke | 9 (11.3) | 2 (10.0) | – |
| Intracerebral hemorrhage | 6 (7.5) | 4 (20.0) | – |
| Traumatic brain injury | 4 (5.0) | 1 (5.0) | – |
| Subarachnoid hemorrhage | 9 (11.3) | 1 (5.0) | |
| Meningitis | 1 (1.3) | – | – |
| Unknown | – | 1 (5.0) | – |
| **Biomarkers, mean (SD)** | | | |
| **At time of clinical assessment** | | | |
| ALAT (U/L) | 154.61 (230) | 128.2 (195) | – |
| Creatinine (μmol/L) | 119.1 (117) | 98 (99) | – |
| Hemoglobin (mmol/L) | 7.0 (1.2) | 7.8 (1.2) | – |
| Amylase (U/L) | 112 (103) | 136.4 (143) | – |
| **At time of last DSA** | | | |
| ALAT (U/L) | – | 127.7 (268) | – |
| Creatinine (μmol/L) | – | 94.3 (110) | – |
| Hemoglobin (mmol/L) | – | 6.8 (1.3) | – |
| Amylase (U/L) | – | 89.6 (55) | - |
| **Inotropic drugs at clinical examination, *n* (%)** | 65 (81.2) | 9 (45.0) | <0.001 |
| **Neuroimaging, total *n*, scan pr pt (SD)** | | | |
| Brain CT | 185, 2.3 (1.7) | 40, 2.0 (2.1) | – |

|  | 1 DSA (*n* = 80) | ≥2 DSA (*n* = 20) | Significant *p*-value |
|---|---|---|---|
| Brain MRI | 4, 0.05 (0.22) | 1, 0.1 (0.22) | – |
| DSA | 80, 1.0 (0.0) | 50, 2.5 (0.69) | <0.001 |
| **Minutes from BD to last DSA, mean (SD)** | 211.2 (260.6) | 2422.4 (1160.2) | <0.001 |
| **Minutes between DSA, mean** | – | 945.7 | – |
| **Indication DSA, *n* (%)** |  |  |  |
| Non-structural brain damage | 61 (76.3) | 15 (75.0) | – |
| Incomplete BD/DNC examination | 8 (10.0) | 1 (5.0) | – |
| Primary lesion infratentorial | 7 (8.8) | 3 (15.0) | – |
| Other | 9 (11.3) | 1 (5.0) | – |
| **Interpretation of first DSA** |  |  |  |
| Sparse or delayed filling, *n* (%) | 8 (10) | 12 (60) | <0.001 |
| **Organ donation** |  |  |  |
| **Consent all organs, *n* (%)** | 80 (100) | 20 (100) | – |
| Actual donor*** | 4 (5.0) | 0 | – |
| Eligible donor# | 0 | 1 (5.0) | – |
| No donor## | 11 (12.5) | 7 (35.0) | 0.01 |
| **Utilized donor###, total number** | 65 (81.3) | 12 (60.0) | 0.04 |
| Heart | 23 | 2 | – |
| Kidneys | 116 | 22 | – |
| Liver | 40 | 6 | – |
| Lung | 22 | 4 | – |
| Pancreas | 7 | 0 | – |
| **Number donated organs, total, mean (SD)** | 206, 2.6 (1.7) | 34, 1.7 (1.6) | 0.03 |

Notes:
* Epilepsy, migraine, ADHD, autism.
** Hypertrophic cardiomyopathy, paroxysmal atrial fibrillation, heart failure, alcohol abuse, cannabis abuse, periodic depression, bipolar disorder, anemia, and generalized myopathy (unspecified).
*** "A donor in whom a surgical incision was made with the intent of organ transplantation or whom at least one organ was retrieved for the purpose of transplantation" (definitions by Scandiatransplant (*Jørgensen & Weinreich, 2023*)).
# "A (medically) suitable person with consent to donation who has been declared dead based on (BD/DNC) criteria [.]" (*Jørgensen & Weinreich, 2023*).
## "[.] a potential donor (who did not become) an eligible, actual, or utilized donor" (*Jørgensen & Weinreich, 2023*).
### "An actual donor from whom at least one solid organ was transplanted" (*Jørgensen & Weinreich, 2023*).

more organs (*n* = 206 organs; mean = 2.6 ± 1.7 SD organs per patient) than patients with ≥2 DSA (*n* = 34 organs; mean = 1.7 ± 1.56 SD organs per patient, *p* = 0.03) (Table 1). Kidneys, livers, and hearts were the most frequently donated organs in both groups. Levels of creatinine and alanine transaminase at the time of DSA were not different between the two groups (Table 1).

## Residual intracranial circulation and blood pressure support

Seventy-four patients required inotropic drug administration at time of clinical BD/DNC evaluation, and 26 patients did not require inotropic treatment. Baseline characteristic showed no significant difference in age, sex, or indication for DSA (Table 2). The most frequent inotropic drug was noradrenaline, (98.6%) and mean dose of noradrenaline administered was 0.21 mcg/kg/min with a SD 0.26. Fourteen of 74 (18.9%) patients

**Table 2 Clinical characteristics of patients, according to need for inotropic blood pressure support.**

| | Inotropic support* (n = 74) | No inotropic support** (n = 26) | Significant p-value* |
|---|---|---|---|
| Age, mean +/− SD | 41.9 +/− 18.3 | 46.0 +/− 20.7 | – |
| Sex, n (%) | | | |
| Female | 32 (43.3) | 14 (53.8) | – |
| Male | 42 (56.7) | 12 (46.2) | – |
| Indication for DSA, n (%) | | | |
| Non-structural brain damage | 56 (75.6) | 19 (73.1) | – |
| Incomplete BD/DNC examination | 6 (8.1) | 3 (11.5) | – |
| Primary infratentorial lesion | 7 (9.5) | 4 (15.4) | – |
| Other | 8 (10.8) | 2 (7.7) | – |
| Neuroimaging, mean +/− SD | | | |
| CT brain (total number of scans, scan per patient (SD)) | 172, 2.3 (1.9) | 50, 1.9 (1.2) | – |
| MR brain ((total number of scans, scan per patient (SD)) | 3, 0.04 (0.2) | 2, 0.08 (0.3) | - |
| DSA (total number of scans, scan per patient (SD)) | 85, 1.15 (0.4) | 45, 1.73 (1.0) | 0.0001 |
| Inotropic drug support, number of patients (n), (%) standard deviation (SD) | | | |
| Noradrenaline, n | 73 (98.6) | – | – |
| Dose (μg/kg/min), mean SD | 0.22 +/− 0.26 | – | – |
| Adrenaline, n | 3 (4.1) | – | – |
| Dose (μg/kg/min), mean SD | 0.32 +/− 0.17 | – | – |
| Dopamine, n | 2 (2.7) | – | – |
| Dose (μg/kg/min), mean SD | 0.59 +/− 0.30 | – | – |
| Presence of intracranial circulation at first DSA, n (%) | 14 (18.9) | 13 (50.0) | 0.002 |
| Internal carotid artery (segment), median | 4 | 6 | 0.025 |
| Vertebral/basilar artery (segment), median | 3 | 3.5 | 0.14 |

Notes:
* Inotropic drugs administered at the time of the first DSA.
** No inotropic drugs administered at the time of the first DSA.

requiring inotropic drugs had residual intracranial circulation, compared to 13/26 (50%) of the patients without inotropic drugs (OR = 0.23 (95% CI [0.09–0.61]), $p$ = 0.002).

Patients without inotropic support showed more distal contrast filling compared to patients who required inotropic drugs. The median of the most distal contrast filling corresponded to the internal carotid artery segment 6 (C6) and the vertebral artery segments 3 and 4 (V3-V4) in patients without inotropic drugs. By contrast, in patients on inotropic support the contrast material only reached C4 and V3 respectively (Fig. 2). There was a significant difference in the median contrast filling of the internal carotid artery ($p$ = 0.025) between the two groups, but not in median contrast filling of the vertebral artery ($p$ = 0.1402) (Table 2).

## Interrater variability of DSA interpretation

Of the 102 patients with available DSA ($n$ = 100 patients for the primary outcome measure, $n$ = 2 patients with DSA and revoked family consent), we excluded two patients without

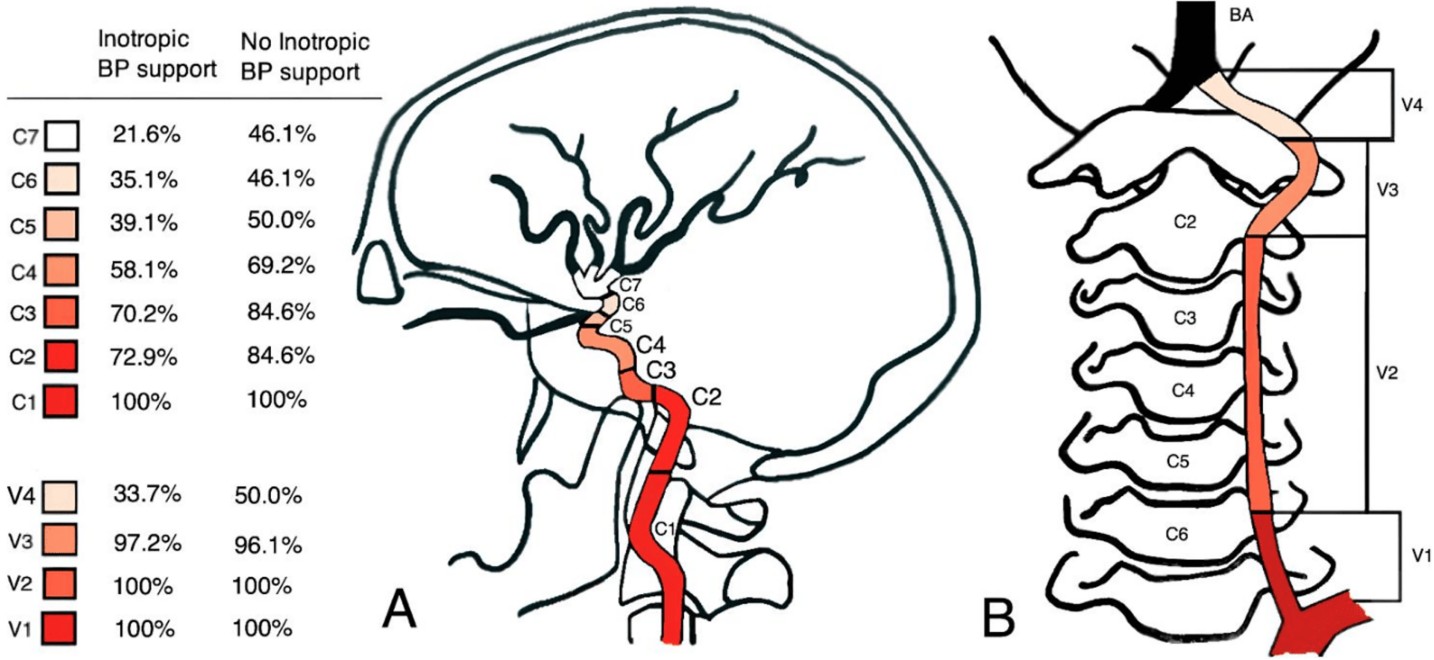

| | Inotropic BP support | No Inotropic BP support |
|---|---|---|
| C7 | 21.6% | 46.1% |
| C6 | 35.1% | 46.1% |
| C5 | 39.1% | 50.0% |
| C4 | 58.1% | 69.2% |
| C3 | 70.2% | 84.6% |
| C2 | 72.9% | 84.6% |
| C1 | 100% | 100% |
| V4 | 33.7% | 50.0% |
| V3 | 97.2% | 96.1% |
| V2 | 100% | 100% |
| V1 | 100% | 100% |

**Figure 2 Schematic overview of DSA results.** The figure depicts the radiological contrast filling degree of the internal carotid arteries (A) and vertebral arteries (B) in patients receiving inotropic drugs ($n = 74$) compared to patients with self-maintained blood pressure ($n = 26$), based on the original radiological report.

**Table 3 Interrater variability of DSA interpretation.**

| | Rater1: 'Intracranial circulation present' | Rater1: 'Intracranial circulation not present' | Total |
|---|---|---|---|
| Rater2: 'Intracranial circulation present' | 43 | 10 | 53 |
| Rater2: 'Intracranial circulation not present' | 5 | 72 | 77 |
| Total | 48 | 82 | 130 |

Note:
2 × 2 table showing interrater variability of DSA interpretation; 88.5% agreement, Cohen's k 0.76; ICC, intracranial circulation.

full radiological DSA report, leaving 100 patients with a total of 130 DSAs for analysis of interrater variability. Interrater agreement was 88.5%, Cohen's kappa = 0.76 (Table 3).

## DISCUSSION

This study's key finding is that patients undergoing repeated DSA during BD/DNC protocols become organ donors less often and donate fewer organs than patients undergoing a single DSA. Furthermore, self-maintained blood pressure at the time of clinical BN/DNC is indicative of residual intracranial circulation on confirmatory DSA, and the radiological interpretation of this procedure is subject to interrater variability. Thus, the more uncertainty about ancillary testing, the higher the likelihood of time delay and the lower the number of donated organs. Given the shortage of organs and ever-increasing transplantation waiting lists, the present results indicate an important area for potential improvements in organ transplantation procedures.

## Delays caused by repeat confirmatory DSA decrease the frequency of organ donation

Patients undergoing more than one confirmatory DSA donate fewer organs. The numbers of lost organs reflect the effects of the time delay until absent intracranial circulation is diagnosed using DSA. This corroborates the notion that time has a negative effect on donor status, highlighting the need to carefully chose and interpret ancillary investigations during BD/DNC determination. In a previous study, the duration of the BD/DNC protocol was associated with negative effects on donations, as increasing time intervals created the potential for confusion about the diagnosis and lead to withdrawal of consent by patient families (*Grzonka et al., 2023*). In another study, no correlation was seen between the consent to organ donation and the time from ICU admission until the completion of the BD/DNC protocol (*Sayan, 2020*). In our cohort, however, family consent was withdrawn in only two patients (who was excluded from the final analysis), so we attribute the declining rate of organ donation in patients investigated with ≥2 DSA to the progressive deterioration of their organs. Although this was not reflected in our study by conventional kidney and liver function blood tests (probably due to the low study power in this regard), organ deterioration is convincingly demonstrated by the fact that patients with repeat DSA donated almost one less organ on average. This underscores the importance of optimal donor management and of avoiding time delays whenever possible (*Floerchinger, Oberhuber & Tullius, 2012*).

## Self-maintained blood pressure predicts residual intracranial circulation

We have shown that absence of inotropic blood pressure support is associated with increased risk of DSA showing residual intracranial circulation. Twenty-six percent of the patients had self-maintained blood pressures. These patients were significantly more likely to have residual intracranial circulation compared to patients on inotropic drugs. This supports a recent study suggesting decreased cerebral perfusion reflects advanced brain herniation (*Salih et al., 2016*). However, negative cerebral perfusion pressure in that study was neither sufficient nor a perquisite for BD/DNC. The authors speculated that in BD/DNC cases with positive CPP arterial blood pressure dropped below a critical closing pressure, thereby causing cessation of cerebral blood flow (*Salih et al., 2016*). *Palmer & Bader (2005)* proposed another hypothetical BD/DNC mechanism in which the intracranial pressure does not exceed the mean arterial pressure: Cerebral blood flow is maintained even though blood flow is too compromised, leading to neuronal damage and eventually BD/DNC. This may constitute the basis for a 'false negative' interpretation of cerebral blood flow studies like DSA.

In any case, self-maintained blood pressure indicates continued functioning of vasomotor centers of the caudal medulla and cervical spinal cord (*Ghali, 2017*). Importantly, most protocols, including the one required by Danish law, do not require inotropic support of blood pressure as a prerequisite for BD/DNC. A subgroup of patients fulfilling clinical BD/DNC criteria can hence still maintain their own blood pressures even though the caudal ventrolateral medulla belongs to the brainstem (*Ghali, 2017*). Similarly,

if polyuria has not yet occurred, this suggests preserved anti-diuretic hormone production by the hypophysis. In these situations, it should come as no surprise that DSA can show weak filling of the proximal intracranial vessels. It has thus been argued that confirmatory tests often lead to confusing results and should be reserved for situations where complete examination of brainstem reflexes is impossible or in primarily infratentorial brain lesions (*Wijdicks, 2011*; *Walter et al., 2018*). This is an important ethical dilemma because the amount of time elapsed is positively correlated with the certainty of lost brain function but negatively with the tissue quality of organ transplants and hence the survival of patients on organ waiting lists (*Rabinstein et al., 2012*; *Messer et al., 2017*).

## Radiological DSA interpretation is subject to interrater variability

The loss of organs transplants after repeated confirmatory investigation is particularly concerning given the suboptimal interrater variability of radiological DSA interpretation (Cohen's kappa 0.76). Even though DSA is considered the gold standard for confirmation of BD/DNC (*Haussmann & Yilmaz, 2020*), the interrater variability in the present study was worse compared to that in a study by *MacDonald, Stewart-Perrin & Shankar (2018)* who investigated a range of confirmatory neuroimaging modalities. In the latter study, a total of 74 patients underwent 41 CT perfusion, 54 CT angiography, 15 radionuclide scans, and 1 DSA. Interrater agreement for CT perfusion, radionuclide scan, and CT angiography was excellent (kappa = 1) (*MacDonald, Stewart-Perrin & Shankar, 2018*). The suboptimal interrater variability in our study is likely related to varying interpretation of the radiologic criteria that state that contrast flow beyond the origin of the ophthalmic artery indicates residual intracranial circulation (*Vatne, Nakstad & Lundar, 1985*). Indeed, the radiological criteria are not as unambiguous as they appear to be: Sometimes faint radiological signal can be seen after a long latency (up to 48 s in one study (*Sawicki et al., 2015*)); we found delayed or sluggish filling in 60% of our patients with residual circulation on first DSA. However, we acknowledge that reevaluation by just one blinded radiologist is not sufficient to make conclusions about how DSA performs compared to other neuroimaging methods (we considered this to be outside the scope of this study), but at least we could document that uncertainty about whether the radiological criteria are fulfilled triggers a repeated DSA with negative effects on organ donation rates. Importantly, the optimal time point when a repeat DSA should be performed is unknown. There is also no consensus about this time point, neither in local Danish protocols nor, as far as we know, internationally. In our study, the average interval between DSAs was almost 16 h—enough time for many organs to become unsuitable for donation.

## Controversies related to confirmatory testing in BD/DNC determination

There remains controversy about the use of ancillary testing in BD/DNC determination (*Joffe, Khaira & de Caen, 2021*). On one hand, it has been argued that ancillary tests are unnecessary because BD/DNC is a clinical state that cannot be confirmed by imaging tests (*Egea-Guerrero et al., 2011*). It has also been stated that these tests may create confusion, in particular when multiple ancillary tests are performed (of note, according to Wijdicks,

'unsupported blood pressures with no need for vasopressors should also be a reason to give pause' during BD/DNC determination (*Wijdicks, 2013*)). On the other hand, it has been reasoned that ancillary radiological studies should always be done to rule out confounding conditions which may erode public trust (*Roberts et al., 2010*; *Lewis & Greer, 2017*). A recent Canadian cross-section study shows most respondents agree that an ancillary test be conducted at least when a complete clinical evaluation is impossible, and that this test should be based on cerebral blood flow studies using DSA, nuclear imaging, CT angiography, and/or CT perfusion (*Chassé et al., 2022*; *Chakraborty & Dhanani, 2017*; *Taylor et al., 2014*; *Garrett et al., 2018*; *Smit et al., 2012*). The authors argue that the requirement for high diagnostic specificity and the common occurrence of confounding factors suggest that the use of clinical criteria and judgment alone is problematic (*Lewis & Greer, 2017*; *Joffe, Hansen & Tibballs, 2021*). In the same vein, *Walter et al. (2018)* have reasoned that brainstem lesions theoretically may cause an 'apneic total locked-in syndrome', in which cerebral blood flow and (rudiment) EEG activity may be present, which is why the demonstration of 'either ancillary finding, electro-cortical inactivity or, preferably, cerebral circulatory arrest, is mandatory for diagnosing BD in patients with a primary infratentorial brain lesion'. However, as stated, Danish legislation requires confirmatory DSA in BD/DNC due to primary infratentorial lesions, so the present data here show that even when adhering to the arguments put forward by *Walter et al. (2018)* and others, all controversies are not entirely resolved.

A metanalysis concluded that because of a high risk of bias and inadequate statistical precision, there is insufficient evidence to determine if any ancillary tests accurately identify BD/DNC (*Wijdicks et al., 2010*). Several studies revealed the presence of brain perfusion, despite absence of clinically detectable brain function on BD/DNC determination (*Flowers & Patel, 2000*). It has been suggested that during the acute phase after the injury, increased intracranial pressure prevents cerebral blood flow, but with the resolution of brain edema, blood flow may be restored to some extent despite irreversible brain damage (*Schröder, 1983*). This, however, would require that a sufficiently long time (days-weeks) has elapsed since clinical BD/DNC and the ancillary test, which was not the case in our cohort. In our opinion, the more likely explanation is that the transition from (rudimentarily preserved) cerebral activity to (a clinical diagnosis of) BD/DNC and further to the (indirectly confirmed) death of the last cerebral neuron is not sharply defined but typically occurs over several hours. As the clinical examination relies on the assessment of the brainstem, it should not come as a surprise that isolated reports exist of islands of electrically functioning cortical neurons giving rise to residual EEG activity (*Grigg et al., 1987*). Similarly, DSA may reveal rudimentary intracranial perfusion for a limited time after a correctly performed clinical BD/DNC examination.

## Strengths and limitations

To our knowledge, this study is the largest evaluating DSA after BD/DNC and the first to assess the effect of repeat DSA on the frequency of organ donation. Strengths include the full case ascertainment over a 7-year period from a large tertiary referral center. This makes the study sample representative of the population of interest and strengthens

external validity. However, due to the retrospective nature of this study, we were unable to assess a sufficient array of blood markers to evaluate organ quality directly. Multivariable comparisons were not performed due to the limited sample size of our study. Cohort studies like this can only establish associations between exposure and outcome, but not prove causality. Finally, our DSA data cannot be applied to non-invasive confirmatory neuroimaging studies.

## CONCLUSIONS

More organs are being donated by patients undergoing a single DSA compared to patients with ≥2 DSA, indicating that delays in BD/DNC protocols owing to repeated confirmatory testing have negative effects on organ quality and result in lost donors and organs. This is concerning since DSA interpretation may be subject to substantial interrater variability. A clinical pointer toward residual intracranial circulation is the presence of self-maintained blood pressure at the time of clinical BD/DNC exam: Patients without the need for inotropic medication are more likely to have residual intracranial circulation. Currently, BD/DNC protocols do not consider whether patients maintain their blood pressure or require inotropic support. Thus, residual intracranial circulation must also be expected (but remains undetected) in many organ donors for whom ancillary testing is not required. In any case, confirmatory testing is an important ethical dilemma because the amount of time elapsed is positively correlated with the certainty of lost brain function but negatively with the tissue quality of organ transplants. Fortunately, the World Brain Death Project is beginning to address some of these controversies (*Greer et al., 2020*). In our opinion, to increase the rate of organs donated, more evidence is required to identify when precisely faint traces of residual intracranial circulation during confirmatory blood flow studies like DSA can safely be disregarded.

### Funding

This work was funded by Offerfonden (https://civilstyrelsen.dk/sagsomraader/raadet-for-offerfonden). Material execution, content and results are the sole responsibility of the authors. The assessments and views expressed in the material are the authors' own and are not necessarily shared by the Offerfonden. Additional funding was supplied by Region Hovedstadens Forskningsfond (https://www.regionh.dk/til-fagfolk/Forskning-og-innovation/finansiering-og-fonde/s%C3%B8g-regionale-midler/Sider/Region-Hovedstadens-forskningsmidler.aspx), the Lundbeck Foundation and Rigshospitalets Forskningspuljer (https://www.forskningspuljer-rh.dk/). The funders had no role in study design, data collection and analysis, decision to publish, or preparation of the manuscript.

### Grant Disclosures

The following grant information was disclosed by the authors:
Offerfonden.

Region Hovedstadens Forskningsfond.
Lundbeck Foundation and Rigshospitalets Forskningspuljer.

## Competing Interests

The authors declare that they have no competing interests.

## Author Contributions

- Karen Irgens Tanderup Hansen performed the experiments, analyzed the data, prepared figures and/or tables, authored or reviewed drafts of the article, and approved the final draft.
- Jesper Kelsen conceived and designed the experiments, performed the experiments, analyzed the data, prepared figures and/or tables, authored or reviewed drafts of the article, and approved the final draft.
- Marwan H. Othman performed the experiments, analyzed the data, authored or reviewed drafts of the article, and approved the final draft.
- Trine Stavngaard performed the experiments, analyzed the data, authored or reviewed drafts of the article, and approved the final draft.
- Daniel Kondziella conceived and designed the experiments, performed the experiments, analyzed the data, prepared figures and/or tables, authored or reviewed drafts of the article, and approved the final draft.

## Human Ethics

The following information was supplied relating to ethical approvals (*i.e.*, approving body and any reference numbers):

The study was approved by the National Committees of Scientific Ethics in Denmark (journal number: 2213160).

## Data Availability

The raw measurements are available in the Supplemental File.

## Supplemental Information

Supplemental information for this article can be found online at http://dx.doi.org/10.7717/peerj.15759#supplemental-information.

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
