# Peer review of "Confirmatory digital subtraction angiography after clinical brain death/death by neurological criteria: impact on number of donors and organ transplants"

_PeerJ, doi:10.7717/peerj.15759_

## Round 0.1 · original submission · Major Revisions

· Academic Editor

Major Revisions

Major revisions are needed. Please do the revisions as requested.

Reviewer 1 ·

Basic reporting

1. The authors provided a clear and comprehensive description of the study design and outlined the recruitment process and inclusion criteria for participants, which were well thought out and justified based on the aims of the study.

2. A mix of Arabic numerals and English numerals is used in the text. Recommend to choose one numeral system and stick to it consistently. For example, in line 222, twenty-six %.

3. In line 184, replace 13/16 with 13/26.

Experimental design

1. The statistical analysis section is well-described and it also included detailed information about the results and methods used.

2. The primary objective was to evaluate the effects of the time delay owing to repeated confirmatory DSAs on the frequency of organ donation. In the results section, more patients with a single DSA were found to become donors than patients with more than 1 DSA. It will be interesting to find the relationship between mean time from clinical brain BD/DNC evaluation until the last DSA and the denotation selection to better understand how longer the delay will affect organ donation.

Validity of the findings

1. In the discussion section, line 246-264, the interrater variability of radiological DSA interpretation is compared with another study, which mainly uses CT. Given the better performance of CT (higher interrater agreement), can CT be recommended as ancillary tests of choice for BD/DNC diagnosis procedures? More discussions can be added.

·

Basic reporting

Basic reporting
English used is good.
Introduction is good but best to include the whole criteria that makes it easier for readers to understand why 2 DSAs are needed right from the beginning.
Structure is good

Experimental design

When should the second DSA be performed/criteria specified?
Re-reported by a single neurointerventionalist, why only one?
What is the criteria that requires 2 DSAs? Is it uncertainty of the first DSA? (line 263)
Best to include the guidelines for BD/DNC in English for Denmark. The reference 12 is not in English and I am unable to confirm statements from line 86-92.
Why is it that the delay caused by 2 DSAs results in less organs procured? Are there other confounding factors other than duration of time and was there objective criteria to decide when the organ could not be used? Who decides when organs cannot be procured?

What was the target blood pressure MAP/CPP for those on inotropic support? I would believe that the MAP would be a more important determinant than dose of inotrope. A patient with MAP of 110mmHg is more likely to have intracranial circulation compared to a patient with MAP of 60mmHg during the time of DSA. Is it more likely that patients that need inotropic support already suffered cardiopulmonary center/brainstem ischemia?

Validity of the findings

Are there other papers that have done interrater variability studies for DSA for the purpose of intracranial circulation? The paper that was cited only did one DSA. (line 255). I strongly agree that only using one radiologist is insufficient to come to the conclusion made as there could be a strong bias in view point and the radiologist likely knows the reason he or she is reading the DSAs again. Anyhow, the criteria of no flow beyond the opthalmic artery should be objective and simple enough. I am not certain, how there could be much variability in interpreting this.
For the CPP to be below the critical closing pressure, it is assuming that all cases of brain death result in increase in ICP. It is however possible for insults to occur only at the brainstem causing braindeath, without an increase in ICP during the early phases.

The main contention in this article is about the need for repeat DSAs but the criteria when this is needed is not stated clearly since it seems that many cases only needed one DSA. There are 2 main problems in this article:
1. The delay in obtaining confirmation seems to be in the lack of objective criteria in determining when intracranial circulation is no longer present and possibly the lack of training or standardization amongst radiologist. There should also be acceptance of other ancillary testing other than just DSAs to avoid this dillema.
2. How is it that self-maintained blood pressure creates diagnostic uncertainty? Do all patients with self-maintained blood pressure require DSA? Of course patients with self-maintained blood pressure would more likely have residual intracranial circulation as the likelihood of brainstem death would be lower since the cardiac/vasomotor centers are still intact and thus the need for ancillary testing in these patients if brainstem reflexes cannot be adequately tested. The message from this conclusion is contradictory.

Additional comments

Overall, the article is good.
However, the message being sent is not clearly backed by what was investigated and is confusing.
It is common sense that the more DSAs done, more time is wasted and therefore less organ transplants.
Poor interrater variability does not mean 2 DSAs are not needed. It may be needed if the first test is not confirmatory. It means other ancillary tests are needed or better training or better objective guidelines.
It is not clear how self-maintained blood pressure has anything to do with delays in brain death diagnosis.

---

## Round 0.2 · accepted · Accept

· Academic Editor

Accept

Thank you. Your manuscript is now ready to go forward to be processed for publication.

Reviewer 1 ·

Basic reporting

I think that the authors have adequately addressed the comments made by the reviewers in the revised version of the manuscript. Therefore, I have no further comments.

Experimental design

.

Validity of the findings

.

·

Basic reporting

Changes made has addressed the concern on this part.

Experimental design

It would have been better if the uncertainty of DSAs was addressed more thoroughly. However this would be an acceptable limitation.

Validity of the findings

Changes are adequate for a clearer understanding of the subject